# Text2Interaction: Establishing Safe and Preferable Human-Robot Interaction

**Jakob Thumm**
Technical University of Munich
`jakob.thumm@tum.de`

**Christopher Agia**
Stanford University
`cagia@stanford.edu`

**Marco Pavone**
Stanford University
`pavone@stanford.edu`

**Matthias Althoff**
Technical University of Munich
`althoff@tum.de`

**Abstract:** Adjusting robot behavior to human preferences can require intensive human feedback, preventing quick adaptation to new users and changing circumstances. Moreover, current approaches typically treat user preferences as a reward, which requires a manual balance between task success and user satisfaction. To integrate new user preferences in a zero-shot manner, our proposed Text2Interaction framework invokes large language models to generate a task plan, motion preferences as Python code, and parameters of a safety controller. By maximizing the combined probability of task completion and user satisfaction instead of a weighted sum of rewards, we can reliably find plans that fulfill both requirements. We find that $83\,\%$ of users working with Text2Interaction agree that it integrates their preferences into the plan of the robot, and $94\,\%$ prefer Text2Interaction over the baseline. Our ablation study shows that Text2Interaction aligns better with unseen preferences than other baselines while maintaining a high success rate. Real-world demonstrations and code are made available at [sites.google.com/view/text2interaction](sites.google.com/view/text2interaction).

**Keywords:** Human-Robot Interaction, Human Preference Learning, Task and Motion Planning, Safe Control.

## 1 Introduction

We are moving toward a future where robots are fully integrated into our everyday lives, from manufacturing [1] to healthcare [2] to households [3]. In these settings, robots must quickly adapt to individual human preferences and changing circumstances. However, current robotic platforms severely lack in this aspect, which prevents their application to daily tasks and widespread acceptance. Recent works [4–6] present promising approaches to seamlessly incorporate task-level preferences, answering the question, "*What* should the robot do?" In human-robot interaction (HRI), we must further address two key user preferences about the behavior of the robot: motion preferences, which determine "Which *path* should the robot choose?", and control preferences, which dictate "How *fast*, *soft*, or *precise* should the robot be?" Recently proposed methods in motion planning [7–9] and safe control [10–12] tend to require labor-intensive human feedback to adapt to such preferences. Hence, these solutions learn a preferable behavior offline and, therefore, fall short when it comes to situational awareness and fast-changing preferences. Additionally, most recent works [9, 13–17] treat human preferences as an additive reward, requiring them to carefully balance task success and user satisfaction.

In this work, we propose Text2Interaction, a framework to incorporate preferences from a single user instruction in three levels of the robot software stack: task planning, motion planning, and control. As exemplified in Fig. 1, we react to user preferences online by querying a large language

8th Conference on Robot Learning (CoRL 2024), Munich, Germany.

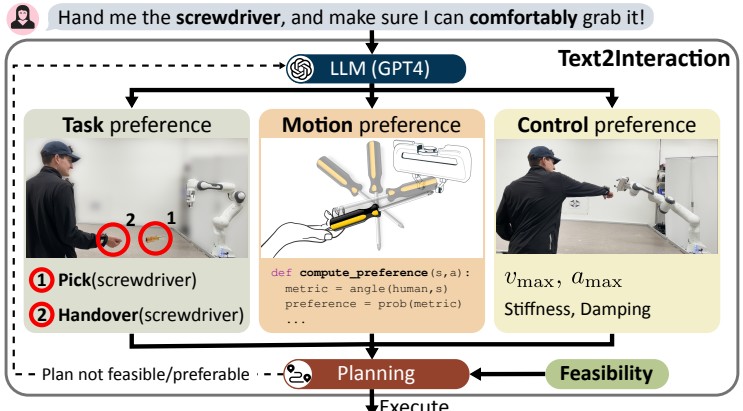

Figure 1: **Preference-aligned planning with Text2Interaction.** The user asks the robot to hand them the screwdriver so that they can comfortably grab it. Text2Interaction queries an LLM to return (a) a sequence of primitives that satisfy the task preferences, (b) a set of motion preference functions as executable Python code, and (c) a set of parameters that adjust the safety controller to the current situation and control preferences of the user. Our planner than aims to find a plan that satisfies the user preferences and is feasible to execute. If the planning step fails, we query the LLM to return the optimal skill for the next timestep only, as discussed in Sec. 6.

model (LLM) to obtain (a) a sequence of primitives as task preference, (b) preference functions written in Python code that evaluate how well an action aligns with the instructed motion preferences, and (c) a set of parameters that adapt our provably safe controller to the control preferences of the user. Our formulation results in the maximization of the combined probability that the plan is feasible and the user is satisfied instead of the commonly used weighted sum of rewards [9, 13–16]. Our ablation study showed that our formulation leads to a robot behavior that is twice as preferable as the baseline while maintaining a high success rate in unseen tasks. Of the 18 participants in our real-world user study, 83 % stated that Text2Interaction considers their preferences, and 94 % preferred Text2Interaction over the baseline [5]. To summarize, our core contributions are:

1. We propose Text2Interaction, a framework for integrating human preferences in robot task planning, motion planning, and control.

2. We derive a task and motion planning formulation that optimizes the likelihood of human satisfaction and task success online.

3. We evaluate Text2Interaction (a) in a user study on a real human-robot collaboration task with 18 participants and (b) in simulation on a set of geometric object rearrangement tasks.

## 2   Related work

Early works in adaptable task planning use a known world model that can be adjusted online to human feedback [18–20]. A number of more recent works propose using LLMs to directly convert language instructions into task plans, either using text [4, 5, 21] or code [22, 23]. Subsequent works attempt to improve the planning robustness of LLMs through iterative re-prompting [24] or by leveraging formal logic representations [25–28], e.g., in the form of a planning domain definition language (PPDL) [29] description. The works in [30–32] extend such a PPDL formulation to collaborative scenarios. Multiple works [14, 33–36] further show how these approaches can allocate tasks more preferably in HRI.

A common way to incorporate human preference on the motion level is to learn a reward function that represents the human intent, typically by asking comparative questions [17, 37–43]. The authors in [12, 44] demonstrated how this form of preference learning can perform preferable robot-human handovers. The main disadvantage of this type of reward learning is that it either only reflects simple

linear reward functions or requires thousands of human queries. As such, several recent works construct reward models with LLMs and use them to train [15, 16, 45–47] or directly synthesize [13, 48] robot skills. Notably, these approaches consider preferences for individual skills, whereas we focus on entire skill sequences. Additionally, these methods generally produce reward functions as linear combinations of reward terms, which requires careful weighing of the reward terms. Closely related to our work is that of Wang et al. [49], which proposes to incorporate human feedback into long-horizon planning by extracting skills from demonstrations and training two models: one to predict the correct sequence of skills and one to predict the parameterization of these skills. However, their approach relies on a labor-intensive offline labeling phase and assumes that the preferences of all future users align with these offline labels.

A broad range of works [10, 11, 50–52] further proposes to learn the parameters of admittance controllers from human feedback to achieve comfortable object handling. Huang et al. [53] demonstrated the effectiveness of this approach in an interactive handover task.

## 3 Notation

We denote a skill as the tuple $\psi = (\phi, a, \alpha, \xi)$. The primitive $\phi$ stems from a library of $K$ primitives $\mathcal{L}^\phi = \{\phi^1, \ldots, \phi^K\}$[1], each of which is parameterized by a set of continuous parameters $a \in \mathcal{A}^k, k \in 1, \ldots, K$ (hereafter referred to as *actions*). The controller $\alpha \in \mathcal{L}^\alpha = \{\alpha^1, \ldots, \alpha^M\}$ with parameters $\xi \in \Xi^m, m \in 1, \ldots, M$ returns the system inputs $\theta = \alpha(\phi^k, a^k, t, \xi)$ to follow the parameterized primitive at time $t$. For our problem, we define a Markov decision process (MDP) with the tuple $M = (\mathcal{S}, \Psi, T, r, \mathcal{S}_0)$. Here, $\mathcal{S}$ is the continuous state space, $\Psi = \mathcal{L}^\phi \times \mathcal{A}^k \times \mathcal{L}^\alpha \times \Xi^m$ is the set of possible skills, $T(s_{t+1} \mid s_t, \psi_t)$ is the transition distribution, and $\mathcal{S}_0 \subseteq \mathcal{S}$ is the set of initial states. The reward function $r : \mathcal{S} \times \Psi \times \mathcal{S} \to \{0, 1\}$ returns $r = 1$ if the execution of a skill is feasible, and $r = 0$ otherwise.

## 4 Problem statement

The user gives an instruction $i$, which may include their task, motion, and control preferences. Thus, we define two events[2] for any environment history $(s_{1:H+1}, \psi_{1:H}) := [s_1, \psi_1, s_2, \psi_2 \ldots, s_{H+1}]$:

$S_{\text{feasible}} : r(s_1, \psi_1, s_2) = \cdots = r(s_H, \psi_H, s_{H+1}) = 1$, i.e., the execution of $\psi_{1:H}$ is feasible,

$S_{\text{preference}} : \psi_{1:H}$ satisfies the user preferences in $i$ when starting in $s_1$.

We assume that a user is satisfied with an environment history if the event $S_{\text{user}} : S_{\text{preference}} \wedge S_{\text{feasible}}$ occurs. Therefore, our goal is to maximize the probability of user satisfaction based on the instruction $i$ and initial state $s_1$ with

$$p(S_{\text{user}} \mid i, s_1, \psi_{1:H}) = p(S_{\text{preference}} \mid i, s_1, \psi_{1:H}, S_{\text{feasible}}) \, p(S_{\text{feasible}} \mid s_1, \psi_{1:H}), \qquad (1)$$

which represents the objective defined in [5, Eq. 2]. Contrary to Lin et al. [5], which only adhere to task preferences, we assume that the user is only satisfied with the execution of a skill sequence $\psi_{1:H}$ if their task, motion, and control preferences are satisfied. Therefore, we define the three events:

$S_{\text{task}} : \phi_{1:H}$ satisfies the task preferences in $i$ when starting in $s_1$,

$S_{\text{motion}} : a_{1:H}$ satisfies the motion preferences in $i$,

$S_{\text{control}} : \alpha_{1:H}, \xi_{1:H}$ satisfy the control preferences in $i$.

Using these events, we can redefine the preference satisfaction event as $S_{\text{preference}} : S_{\text{task}} \wedge S_{\text{control}} \wedge S_{\text{motion}}$, and its probability as

$$p(S_{\text{preference}} \mid i, s_1, \psi_{1:H}, S_{\text{feasible}}) = p(S_{\text{task}}, S_{\text{control}}, S_{\text{motion}} \mid i, s_1, \psi_{1:H}, S_{\text{feasible}}) = \qquad (2)$$

$$p(S_{\text{motion}} \mid i, s_1, \psi_{1:H}, S_{\text{feasible}}, S_{\text{task}}, S_{\text{control}}) \, p(S_{\text{control}} \mid i, s_1, \phi_{1:H}, \alpha_{1:H}, \xi_{1:H}, S_{\text{feasible}}, S_{\text{task}}) \qquad (3)$$

$$p(S_{\text{task}} \mid i, s_1, \phi_{1:H}, S_{\text{feasible}}),$$

---

[1]Note that we denote the $k$-th primitive in the library $\mathcal{L}^\phi$ as $\phi^k$ and the primitive active at time $t$ as $\phi_t$.

[2]All events defined in this work are the "success" event of its Bernoulli trial.

where we assume that $S_{\text{task}}$ is independent of $a_{1:H}$, $\alpha_{1:H}$, and $\xi_{1:H}$ and $S_{\text{control}}$ is independent of $a_{1:H}$. Thus, to find the skill sequence with the highest likelihood of user satisfaction, we must solve

$$\psi_{1:H}^{\star} = \underset{\psi_{1:H}}{\arg\max} \; p(S_{\text{feasible}} \mid s_1, \psi_{1:H}) \, p(S_{\text{motion}} \mid i, s_1, \psi_{1:H}, S_{\text{feasible}}, S_{\text{task}}, S_{\text{control}}) \quad (4)$$
$$p(S_{\text{control}} \mid i, s_1, \phi_{1:H}, \alpha_{1:H}, \xi_{1:H}, S_{\text{feasible}}, S_{\text{task}}) \, p(S_{\text{task}} \mid i, s_1, \phi_{1:H}, S_{\text{feasible}}) \, .$$

The goal of this work is to derive a framework including reasonable assumptions and approximations to efficiently find a solution to (4) that adheres to the given preferences in an HRI setting.

## 5 Supporting methodology

This section introduces the primitive learning method and safety controller used in our work.

**Sequencing task-agnostic policies**  For each primitive $\phi^k \in \mathcal{L}^{\phi}$, we use sequencing task-agnostic policies (STAP) [54] to learn a policy $\pi^k(a \mid s)$ from which one can sample actions for planning and a Q-value function $Q^k \colon \mathcal{S} \times \mathcal{A} \to [0, 1]$ which estimates the probability that executing primitive $\phi^k$ parameterized by action $a$ in state $s$ is feasible. Additionally, we learn the transition distribution $T(s_{t+1} \mid s_t, \psi_t)$ to predict the next state. Given a dataset of transitions $(s_0, a_0, s_1, r_0) \in \mathcal{D}^k = \mathcal{S}_0 \times \mathcal{A}^k \times \mathcal{S} \times \{0, 1\}$, STAP learns the functions $\pi^k$, $Q^k$, and $T$ using offline reinforcement learning. In our experiments, we reduce sample complexity by training these components independently from the controller $\alpha(\cdot, \xi)$, i.e., we assume that the distribution of skill success is approximately constant across most controller configurations: $\forall \alpha^1, \xi^1, \alpha^2, \xi^2 : p(S_{\text{feasible}} \mid s_1, \phi, \alpha^1, \xi^1, a) \approx p(S_{\text{feasible}} \mid s_1, \phi, \alpha^2, \xi^2, a)$. This assumption holds in our experiments as the controller parameters mainly impact the speed with which the robot executes the primitive, but not its path.

**Safety controller**  To ensure human safety, we use a provably safe controller [55–57] that adheres to ISO 10218-2 [58] and ISO/TS 15066 [59]. Our controller comes in three variants: $\alpha_{\text{stop}}$ [57, 60], $\alpha_{\text{contact}}$ [60, 61], and $\alpha_{\text{compliant}}$ [62]. Using $\alpha_{\text{stop}}$, we guarantee that the robot comes to a complete stop before any contact with a human could occur. The $\alpha_{\text{contact}}$ mode allows the robot to have a low speed close to the human and thereby enables active contact. Finally, the $\alpha_{\text{compliant}}$ mode ensures low contact forces between the end-effector and the human using compliant Cartesian control. Generally, we can use the parameter vector $\xi$ to adapt the maximal velocity, acceleration, jerk, stiffness, and damping of the controller. To simplify the control parameter selection for the LLM, we predefine a set of parameter vectors: $\xi_{\text{coexistence}}$ for non-interactive scenarios, $\xi_{\text{critical}}$ for high-risk scenarios, and three parameter vectors for interaction with varying user experience $\xi_{\text{beginner}}$, $\xi_{\text{intermediate}}$, and $\xi_{\text{expert}}$.

## 6 The Text2Interaction framework

This section details how Text2Interaction satisfies the user preferences and achieves feasible plans. To find feasible skill sequences in long-horizon tasks, Lin et al. [5] propose two modes: *shooting* and *greedy-search*. In the following paragraphs, we derive the assumptions and approximations necessary to incorporate motion and control preferences into their approach and solve the problem in (4) efficiently. Fig. 2a summarizes our resulting planning objectives, and Fig. 2b exemplifies how Text2Interaction links the *shooting* and *greedy-search* modes.

**Shooting**  In the *shooting* step, we first let an LLM generate all elements necessary to plan a sequence of skills that fulfills the instruction of the user. For the task and control preferences, the LLM directly generates the sequence of primitives $\phi_{1:H}$ and controller settings $\alpha(\cdot, \xi_{1:H})$. Since these elements are now fixed, our planner cannot influence them. Therefore, we set the probability of satisfying the task and control preferences in (4) to $p(S_{\text{task}} \mid i, s_1, \phi_{1:H}, S_{\text{feasible}}) \approx c_{\text{task}}$ and $p(S_{\text{control}} \mid i, s_1, \phi_{1:H}, \alpha_{1:H}, \xi_{1:H}, S_{\text{feasible}}, S_{\text{task}}) \approx c_{\text{control}}$, with $c_{\text{task}}, c_{\text{control}} \in (0, 1]$. Hereby, we assume that the suggested task plan and controllers have a non-zero probability of satisfying the user. In the next paragraph, we will discuss how we proceed if this assumption fails. To approximate the probability that the execution of the plan is feasible, Lin et al. [5, Eq. 5] have derived that

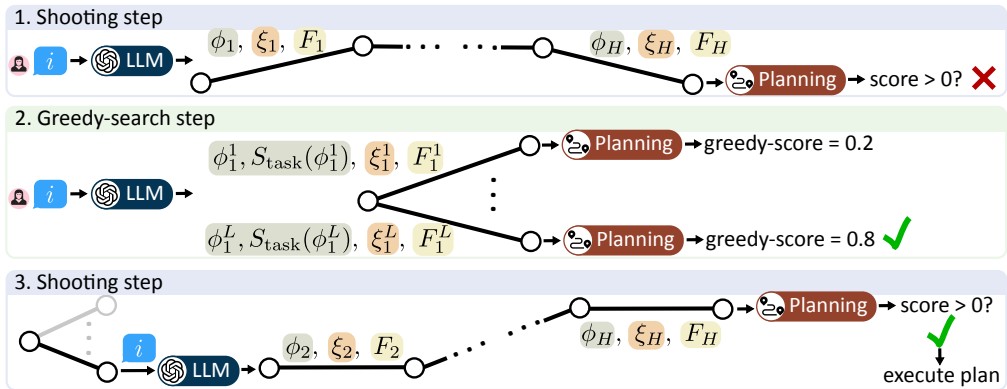

$$\psi_{1:H}^{\star} = \underset{\psi_{1:H}}{\arg\max}\ p(S_{\text{feasible}} \mid s_1, \psi_{1:H})\ p(S_{\text{motion}} \mid i, s_1, \psi_{1:H}, S_{\text{feasible}}, S_{\text{task}}, S_{\text{ctrl}})$$
$$p(S_{\text{control}} \mid i, s_1, \phi_{1:H}, \alpha_{1:H}, \xi_{1:H}, S_{\text{feasible}}, S_{\text{task}})\ p(S_{\text{task}} \mid i, s_1, \phi_{1:H}, S_{\text{feasible}})$$

**Shooting step**

$$a_{t:H}^{\star} = \underset{a_{t:H}}{\arg\max}\ \underbrace{c_{\text{task}}\ c_{\text{control}} \prod_{\tau=t}^{H} F_{\tau}(s_{\tau}, a_{\tau})\ Q_{\tau}(s_{\tau}, a_{\tau})}_{\text{score}}$$

**Greedy-search step**

$$(\phi_t^*, a_t^*) = \underset{\phi_t^l, a_t^l}{\arg\max}\ \underbrace{c_{\text{control}}\ P_{\text{task}}(\phi_t^l)\ F_t^l(s_t, a_t)\ Q_t^l(s_t, a_t)}_{\text{greedy-score}}$$

(a) **Planning objectives** for the shooting and greedy-search step.

(b) **Example of the interplay between shooting and greedy-search**. After receiving an instruction $i$, we first perform a *shooting* step, where an LLM generates a task plan $\phi_{1:H}$, the controller settings $\alpha(\cdot, \xi_{1:H})$, and the preference functions $F_{1:H}$. In this example, the first shooting step failed to find a valid solution. Therefore, we execute a greedy-search step for the first time step, where we generate multiple candidate solutions and select the top candidate. Afterwards, the shooting step starting from $s_2$ succeeds.

Figure 2: Detailed overview of the Text2Interaction framework.

$p(S_{\text{feasible}} \mid s_1, \psi_{1:H}) \approx \prod_{t=1}^{H} Q_t(s_t, a_t)$, where $Q_t$ denotes the $Q$-value function associated with the primitive $\phi_t$ at time $t$. Analogously, we derive in Appendix D that we can approximate the probability that the user is satisfied with the action sequence using a set of preference functions as

$$p(S_{\text{motion}} \mid i, s_1, \psi_{1:H}, S_{\text{feasible}}, S_{\text{task}}, S_{\text{control}}) \approx \prod_{t=1}^{H} F_t(s_t, a_t), \qquad (5)$$

where $s_{t+1} \sim T_t(\cdot \mid s_t, \psi_t)$ for $t \geq 1$. A preference function $F(s, a)$ returns the likelihood that executing action $a$ in state $s$ fulfills the motion preferences of the user. We later discuss how we can construct such functions online and give a number of examples in Appendix C. We can now approximate our objective in (4) and find the optimal sequence of actions $a_{1:H}^{\star}$ that both satisfy the motion preferences $\prod_{t=1}^{H} F_t(s_t, a_t) > 0$ and are feasible $\prod_{t=1}^{H} Q_t(s_t, a_t) > 0$, by using a cross-entropy method planner [63] to solve the optimization problem (see Fig. 2a)

$$a_{1:H}^{\star} = \underset{a_{1:H}}{\arg\max}\ c_{\text{task}}\ c_{\text{control}} \prod_{t=1}^{H} F_t(s_t, a_t)\ Q_t(s_t, a_t). \qquad (6)$$

**Greedy-search** If we cannot find a *shooting* plan that adheres to the human motion preferences and is feasible, i.e., $\prod_{t=1}^{H} F_t(s_t, a_t^{\star})\ Q_t(s_t, a_t^{\star}) = 0$, we have to expect that our assumption that the task sequence is valid $c_{\text{task}} > 0$ failed. In this case, we execute a *greedy-search* [5], where we try to find the optimal next skill $\psi_t^{\star}$ instead of the entire skill sequence $\psi_{1:H}^{\star}$. We define the simplified event $S_{\text{user}}^t = S_{\text{feasible}}^t \wedge S_{\text{task}}^t \wedge S_{\text{motion}}^t \wedge S_{\text{control}}^t$, which occurs if the skill $\psi_t$ is feasible and adheres to the user preferences at time step $t$. Thus, the problem in (4) simplifies to

$$\psi_t^{\star} = \underset{\psi_t}{\arg\max}\ p(S_{\text{feasible}}^t \mid s_t, \psi_t, S_{\text{user}}^{1:t-1})\ p(S_{\text{motion}}^t \mid i, s_t, \psi_t, S_{\text{feasible}}^t, S_{\text{task}}^t, S_{\text{control}}^t, S_{\text{user}}^{1:t-1}) \cdot \qquad (7)$$
$$p(S_{\text{control}}^t \mid i, s_t, \phi_t, \alpha_t, \xi_t, S_{\text{feasible}}^t, S_{\text{task}}^t, S_{\text{user}}^{1:t-1})\ p(S_{\text{task}}^t \mid i, s_t, \phi_t, S_{\text{feasible}}^t, S_{\text{user}}^{1:t-1}),$$

where we made the Markov assumption. In *greedy-search*, we let an LLM return $L \leq K$ candidate primitives for the next skill. We then approximate the probability that the user is satisfied with the next selected primitive with the sum of token log-probabilities of the language description of each primitive $p(S_{\text{task}}^t \mid i, s_t, \phi_t, S_{\text{feasible}}^t, S_{\text{user}}^{1:t-1}) \approx P_{\text{task}}(\phi_t^l)$, $l \in 1, \ldots, L$ as proposed in [4, 5]. These scores represent the likelihood that the textual label of a primitive is a valid next step for the instruction $i$ [4]. We could proceed with the control preferences in the same way, but generating $L^2$ pairs of primitives and control candidates is time-consuming. Thus, we only generate one set of control parameters per candidate primitive and set $p(S_{\text{control}}^t \mid i, s_t, \phi_t, \alpha_t, \xi_t, S_{\text{feasible}}^t, S_{\text{task}}^t, S_{\text{user}}^{1:t-1}) \approx c_{\text{control}}^l$. Then, we generate one preference function per candidate primitive to obtain an approximation of $p(S_{\text{motion}}^t \mid i, s_t, \psi_t, S_{\text{feasible}}^t, S_{\text{task}}^t, S_{\text{control}}^t, S_{\text{user}}^{1:t-1}) \approx F_t^l(s_t, a_t)$, and approximate the probability of feasibility with $p(S_{\text{feasible}}^t \mid s_t, \psi_t) \approx Q_t^l(s_t, a_t)$. Thus, the problem in (7) simplifies to

$$\phi_t^\star, a_t^\star \approx \underset{\phi_t^l,\, a_t}{\arg\max} \quad c_{\text{control}}^l\, P_{\text{task}}(\phi_t^l)\, F_t^l(s_t, a_t)\, Q_t^l(s_t, a_t)\,, \tag{8}$$

which gives us the optimal one-step primitive and action. After finding a solution to the *greedy-search* problem in (8) using a cross-entropy method planner, we return to the regular *shooting* strategy starting from state $s_{t+1}$.

**Generating motion preferences from text**    To approximate the probability that a human would approve action $a_t$ at time $t$, i.e., $p(S_{\text{motion}}^t \mid i, s_t, \psi_{1:H}, S_{\text{feasible}}^t, S_{\text{task}}^t, S_{\text{control}}^t)$, we query an LLM to return a preference function $F_t(s_t, a_t)$ as Python code based on the user instruction $i$, the primitive sequence $\phi_{1:H}$, and the controller $\alpha(\cdot, \xi_{1:H})$. To align the return of the LLM with our desired structure, we prompt it with a system and task description, a number of in-context examples [13], and a set of programmatic building blocks. These building blocks provide the LLM with

1. the predicted next state $s_{t+1} = E_{\bar{s}_{t+1}}\left[T(\bar{s}_{t+1} \mid s_t, \psi_t)\right]$,
2. the functionality to retrieve the pose of an object in a given frame from a given state,
3. functions to calculate metrics given a set of poses, e.g., the Euclidean distance of two objects,
4. a set of monotonic functions $g_p : \mathcal{R} \to [0, 1]$ that map the value of a given metric to a utility.

By restricting the codomain of the utility function to $[0, 1]$, we can interpret its image as the probability of user satisfaction. The LLM can evaluate multiple logical connectives in the preference function probabilistically using the $\text{AND}(p_1, p_2) = p_1 p_2$ and $\text{OR}(p_1, p_2) = p_1 p_2 + p_1(1 - p_2) + p_2(1 - p_1)$ operators, where we assume that the events are stochastically independent. We provide more details about our prompt design in Appendix B. One key assumption that we make is that the preference functions $F_{1:H}(s, a)$ returned by the LLM accurately reflect the true underlying probability $p(S_{\text{motion}} \mid i, s_1, \psi_{1:H}, S_{\text{feasible}}, S_{\text{task}}, S_{\text{control}})$, which we evaluate in our ablation study in Sec. 7.2.

# 7    Experimental evaluation

We performed a user study and an ablation study to investigate three main hypotheses:

**H1**  Users have preferences regarding the motion and control of interactive robots.

**H2**  Text2Interaction integrates these preferences in the execution of the plan of the robot.

**H3**  Text2Interaction generates preference functions that align with the instructions of the user.

Our experiments focused on preferences on the motion and control level, as previous work [4, 5] already investigated task-level preferences extensively.

## 7.1    Real-world user study

To evaluate our research hypotheses, we set up our running example in Fig. 1 on a Franka Research 3 robot. Hereby, the robot had to pick up a screwdriver from the desk and hand it over to the user, as demonstrated in our supplementary video. Our user demographic consisted of 18 participants, of

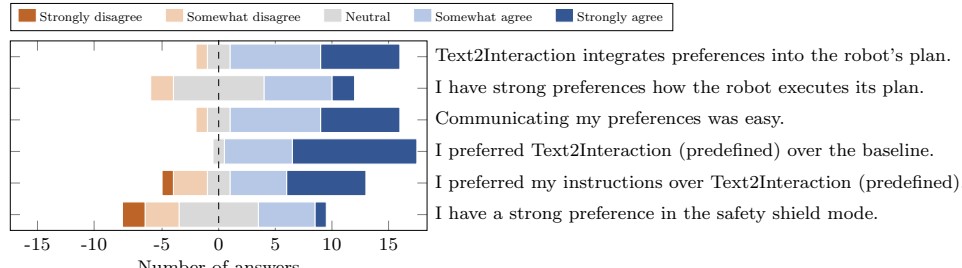

Figure 3: **Main takeaways from our user study.** The answers of the 18 participants are centered around zero for better comparison.

which 16 stated that they had previous experience in robotics. Our experiments then comprised four experimental stages, with the users performing the same task multiple times per stage and answering questions on a five-point Likert scale after each stage.

In *stage 1*, we demonstrated our safety shield to the user. For this, we predefined three sets of controller parameters: $\xi_{\text{beginner}}$, $\xi_{\text{intermediate}}$, and $\xi_{\text{expert}}$, listed in Appendix A, with each mode being faster and more reactive than the previous one. We then let the user decide which mode they would like to work with in the upcoming interactive task.

In *stages 2 and 3*, the users performed the task of the screwdriver handover, where the robot had two different modes. The first mode was baseline 1 [5], which only optimized for task success and did not incorporate motion level preferences. The second mode was Text2Interaction (predefined), which we previously queried with the instruction, "*Hand me the screwdriver, and make sure the handle is pointing towards me so that I can comfortably grab the handle.*".

In *stage 4*, we asked each user for personal instructions to the robot for this task. We then queried Text2Interaction for new custom preference functions and performed the task in a zero-shot manner.

After stages two to four, the users answered five questions about trust, intelligence, cooperativeness, comfort, and awareness [64] to evaluate the quality of the HRI. By asking the same questions repeatedly, we can evaluate if there is a difference in the distribution of answers between the stages using the Wilcoxon signed-rank test [65] and report the $p$-values. Fig. 3 further summarizes the answers to a set of general questions not associated with any specific stage. Our user study confirmed our hypotheses in the following ways:

**H1**: First, users tend to agree that they have a strong preference for the way the robot executes the task, see Fig. 3. Second, with statistical evidence, users perceived Text2Interaction (predefined) as more intelligent ($p \leq 0.005$), more cooperative ($p \leq 0.01$), and more trustworthy ($p \leq 0.05$) than the baseline. Additionally, they were more comfortable with Text2Interaction (predefined) ($p \leq 0.025$) and reported that the robot more accurately perceived what the goals of the users were ($p \leq 0.01$) than with the baseline. Third, from our user group, $50\%$ chose $\xi_{\text{intermediate}}$, and $50\%$ chose $\xi_{\text{expert}}$ as controller parameters, which indicates a general preference towards a faster robot for this task. Furthermore, $33\%$ of users have stated a strong preference for the safety shield mode, indicating that although a majority of users might be indifferent to the controller parameters, it is still relevant to integrate control preferences in HRI.

**H2**: $83\%$ of users agreed that Text2Interaction integrates their preferences into the plan of the robot, and $94\%$ of users preferred the execution of Text2Interaction (predefined) over the baseline. The participants further rated the execution of their personal preferences, Text2Interaction (personal), as more cooperative ($p \leq 0.005$) and comfortable ($p \leq 0.025$) than the baseline and stated that the robot more accurately perceived what their goals are than the baseline ($p \leq 0.025$).

**H3**: $67\%$ of users preferred the robot behavior following their instruction over Text2Interaction (predefined). Additionally, $83\%$ of users stated that it was easy to communicate their preferences with Text2Interaction.

## 7.2 Ablation study

To further quantify the validity of hypothesis **H3**, we performed an ablation study on a set of object rearrangement tasks. Our study aims to validate if Text2Interaction can generate valid preference functions from a limited set of in-context examples. We used 15 tasks consisting of an instruction, a task plan, and hand-crafted oracle preference functions $F^\star_{1:H}$. For each task, we defined three trials, and for each trial, we randomly selected three in-context examples out of the other 14 tasks to construct the LLM prompt[3]. We evaluated each trial 100 times with random initial states and calculated their average preference score based on the oracle preference functions $F^\star_{1:H}$. Finally, we calculated the mean success rates and preference scores together with their $95\%$ confidence intervals using bootstrapping. Our experiments included four agents:

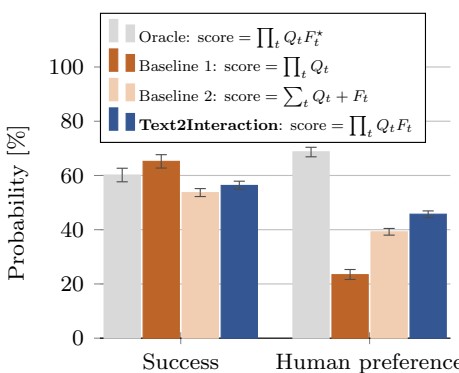

Figure 4: **Mean results of our object arrangement experiments**. The whiskers display the $95\%$ confidence interval in the reported mean metric.

- Oracle, which used the hand-crafted preference functions (score $= \prod_t Q_t F^\star_t$),
- Baseline 1 was [5], which only optimized for task success (score $= \prod_t Q_t$),
- Baseline 2 was inspired by [13, 15, 16], which treats preferences and the feasibility of the plan as rewards, and aimed to maximize the score $= \sum_t Q_t + F_t$,
- Text2Interaction with the objective defined in (6) (score $= \prod_t Q_t F_t$).

Text2Interaction successfully generated executable preference functions for all 45 trials. Fig. 4 shows that our approach achieves approximately the same or higher task success rates than both baselines but achieves significantly higher preference scores. Empirically, Text2Interaction mainly struggled with vaguely communicated preferences, such as, "*Place object A as far left of the table as possible*." This is because there are many different ways of interpreting the instruction, which leads to a natural discrepancy between Text2Interaction and the oracle. Overall, the results of our ablation support hypothesis **H3** and indicate a robust generalization of Text2Interaction to new problems.

## 8 Conclusion and Limitations

We presented a framework to include human preferences from a single user instruction in three levels of the robotic software stack: task, motion, and control. By optimizing over the combined probability that the plan is feasible and satisfies the user instruction, we find task and motion plans that are more likely to fulfill both criteria than related baselines. The overwhelming majority of users found that Text2Interaction integrates their preferences easily.

One limitation of our work is the dependence on in-context examples for the LLM prompts. If the user request is semantically out of distribution of these examples [66], the LLM is unlikely to output reasonable results. Unfortunately, as of now, LLMs tend to hallucinate solutions in such cases instead of returning an empty preference function. Possible solutions could be to perform out-of-distribution detection for in-context examples or fine-tuning an LLM on a large set of preference examples to improve generalization. Furthermore, we assume the user preference is fully communicated through language instructions. Most human-human interactions, however, include non-verbal communication, which Text2Interaction currently does not capture. Future work could integrate other forms of communication through methods like gesture recognition and vision language models.

---

[3]We are using the OpenAI `gpt-4-0125-preview` model with a context length of $128\,000$ tokens. The average query time was $29 \pm 9$ s.

**Acknowledgments**

The authors gratefully acknowledge financial support by the Horizon 2020 EU Framework Project CONCERT under grant 101016007, and by the Deutsche Forschungsgemeinschaft (German Research Foundation) under grant number AL 1185/33-1. This research was funded in part by the Federal Ministry of Education and Research (BMBF), and supported by a fellowship within the IFI programme of the German Academic Exchange Service (DAAD). Toyota Research Institute provided funds to support this work. This work was also supported by the National Aeronautics and Space Administration (NASA) under the University Leadership Initiative (ULI) program.

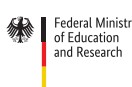
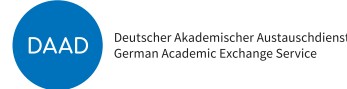

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

# Appendices

## A  Controller parameters

The controller parameters used in our user study are listed in Table 1. We define two different maximal joint acceleration and jerk values. The values $a_\phi$ and $j_\phi$ define how fast the robot accelerates along its primitive. The values $a_{max}$ and $j_{max}$ define how fast the robot is allowed to decelerate in case a human intersects its path. The value $v_{safe}$ defines the maximal Cartesian velocity that a point on the robot geometry is allowed to have upon contact with a human.

Table 1: Controller parameters used in our user study

| Parameter | Description | $\xi_{beginner}$ | $\xi_{intermediate}$ | $\xi_{expert}$ |
|---|---|---|---|---|
| $v_{max}$ | Maximal joint velocity $[\mathrm{rad/s}]$ | 0.25 | 0.4 | 2.0 |
| $a_\phi$ | Maximal joint acceleration on the primitive $[\mathrm{rad/s^2}]$ | 0.5 | 1 | 2.5 |
| $a_{max}$ | Maximal joint acceleration $[\mathrm{rad/s^2}]$ | 2 | 5 | 20 |
| $j_\phi$ | Maximal joint jerk on the primitive $[\mathrm{rad/s^3}]$ | 2 | 5 | 15 |
| $j_{max}$ | Maximal joint jerk $[\mathrm{rad/s^3}]$ | 100 | 200 | 400 |
| $v_{safe}$ | Maximal contact velocity (Cartesian) $[\mathrm{m/s}]$ | 0.05 | 0.15 | 0.2 |

## B  Prompt design

Our prompts contain the following elements, which are further summarized in Fig. 5.

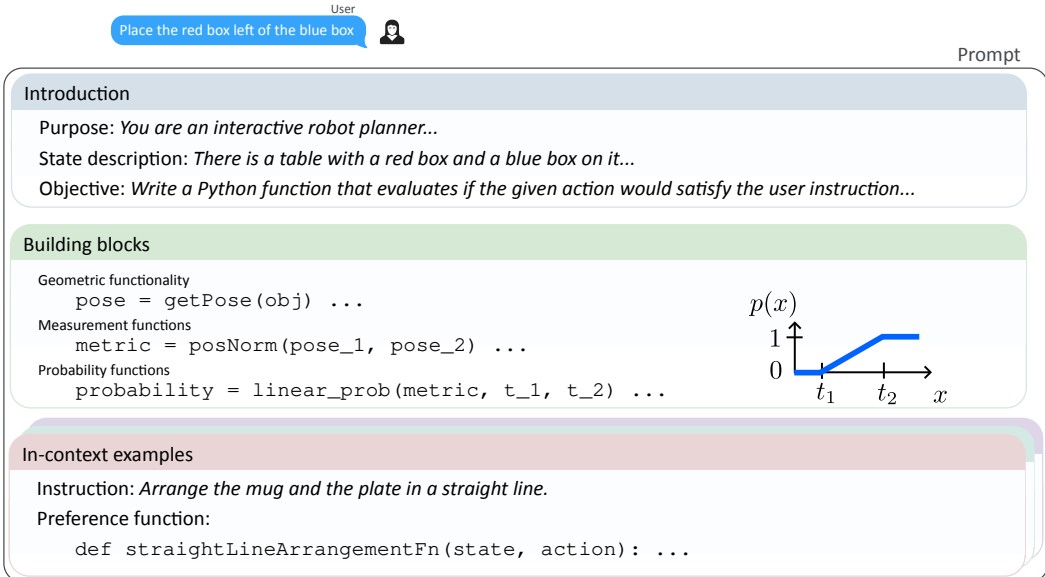

Figure 5: Structure of the prompt used to generate the output of Text2Interaction.

First, we use a fixed system prompt comprising:

- Introduction and purpose of the system.
- Definition of terminology: task plan, primitive, control parameters, and preference function.
- State definition: objects, predicates, and relationships.
- Available manipulation primitives.

- Objective: Structure and constraints of: task sequence, controller parameters, and preference functions.
- Preference function signature in Python code.

Then, we provide a number of building block functions as listed in Fig. 6 that the LLM can use to construct the preference function in Python code.

```
1   # Get the pose of an object in a specified frame from the current
    environment state.
2   pose = getPose(state, obj, frame='world')
3   # Get the L1, L2, or L_inf norm of the positional difference of the
    two poses. Select axis to evaluate the norm on.
4   metric = positionNorm(pose_1, pose_2, norm='L2', axis=['x', 'y', 'z'
    ])
5   # Calculate the difference in rotation of two poses using the great
    circle distance.
6   metric = greatCircleDistance(pose_1, pose_2)
7   # Evaluate if an object is pointing in a given direction. Rotates the
     given main axis by pose_1.orientation and calculates the
    greatCircleDistance between the rotated axis and pose_2.position.
8   metric = pointingInDirectionMetric(pose_1, pose_2, main_axis=[1, 0,
    0])
9   # Calculate the rotational difference between pose_1 and pose_2
    around the given axis.
10  metric = rotationAngle(pose_1, pose_2, axis)
11  # Return 1.0 if metric >= t, 0.0 otherwise. And vice versa if not
    direction.
12  prob = threshold(metric, t, direction=true)
13  # Return 1.0 if metric >= t_2, 0.0 if metric < t_1, and linearly
    interpolate otherwise. And vice versa if not direction.
14  prob = linear(metric, t_1, t_2, direction=true)
15  # Normal cummulative distribution function with given mean and
    standard deviation
16  prob = normal(metric, mean, std_dev, direction=true)
17  prob = AND(prob_1, prob_2)
18  prob = OR(prob_1, prob_2)
19
```

Figure 6: Helper functions defined in our experiments to construct the motion preference function.

A number of in-context examples in the prompt serve as a guideline for the LLM to construct outputs in the correct format. Our in-context examples start with an instruction with the following elements:

- State description: objects in the scene, predicates and relations of the objects to each other.
- Orientation definition: where is front/behind, left/right, up/down.
- User instruction $i$.
- If the task plan is already fixed, like in our experiments, we provide it here.
- If the controller parameters are already fixed, like in our experiments, we provide them here.

Each in-context example defines a hand-scripted solution to its given instruction consisting of a task plan $\phi_{1:H}$, the controller parameters $\xi_{1:H}$, and a sequence of preference functions $F_{1:H}$. Finally, we add the actual instruction to the prompt consisting of the same elements as the example instructions in the in-context examples.

## C   Examples for preference representations

In this section, we provide examples how Text2Interaction integrates human preference effectively.

**Object arrangement** When given instruction $i$: "Place the object left of the blue box", Text2Interaction returns a preference function $F_t(s_t, a_t)$ of the place primitive as exemplified in Fig. 7.

```python
def isLeftOfBlueBlox(state, action):
    # Get the blue box pose in the current state
    blue_box_pose = get_pose(state, "blue_box")
    # Get the predicted object pose after executing the action
    next_state = transition_fn(state, action)
    object_pose = get_pose(next_state, "object")
    # Evaluate if the object is placed left of the blue box
    left = [0.0, 1.0, 0.0]
    diff_left = position_diff_along_direction(object_pose, blue_box_pose, left)
    # The direction difference should be greater zero.
    t_0 = 0.0
    # A distance of 10cm is preferred.
    t_1 = 0.1
    is_left_probability = linear_probability(diff_left, t_0, t_1)
    return is_left_probability
```

Figure 7: Example preference function for the instruction "Place the object left of the blue box".

**Safety preference** In our recurring example of Fig. 1, the user might instruct "I do not want to get hit by the tip of the screwdriver". Text2Interaction could respect this safety concern in two different ways:

1. A restrictive controller is selected, e.g., $\alpha_{\text{stop}}(\cdot, \xi_{\text{critical}})$, which guarantees that the robot comes to a full stop early before a collision could occur. Then, we do not need to account for any discomfort due to collisions in the preference function. For example, we could pick an action, where the tip of the screwdriver faces the human most of the time and only switches directions in the end.

2. A less restrictive controller is selected, e.g., $\alpha_{\text{contact}}(\cdot, \xi_{\text{intermediate}})$. We could then account for the instructed preference on the motion level by requiring all handover actions to have the tip of the screwdriver face away from the human during the entire trajectory.

**Carefulness** Another example, where Text2Interaction has multiple ways of incorporating human preference into the plan of the robot follows the instruction $i$: "Put the vase on the desk carefully!". As an example, Text2Interaction could integrate the instructed preference on two different levels:

1. *On the motion level as a preference function*: We could take the angle between the base of the vase and the desk surface as a metric. A small angle could then relate to a more careful, and therefore preferred, touchdown.

2. *On the control level*: Text2Interaction could choose an admittance controller ($\alpha_{compliant}$) instead of a PID-controller with a low stiffness and low overall low velocity, which would lead to softer touchdowns.

**Comparison to baseline objectives** Our objective in (6) leads to an optimization of the product of $Q$-functions and preference functions. In our ablation study, we compare our approach to the common weighted sum of rewards formulation used in many baselines [13–16]. In Fig. 8, we give an intuition why our formulation outperforms the baselines.

## D   Derivations

In this section, we derive the approximations of (5) under the assumptions of the Markov property, i.e., all information of $s_{1:t-1}, a_{1:t-1}$ is contained in $s_t$. Additionally, we assume that the user has a

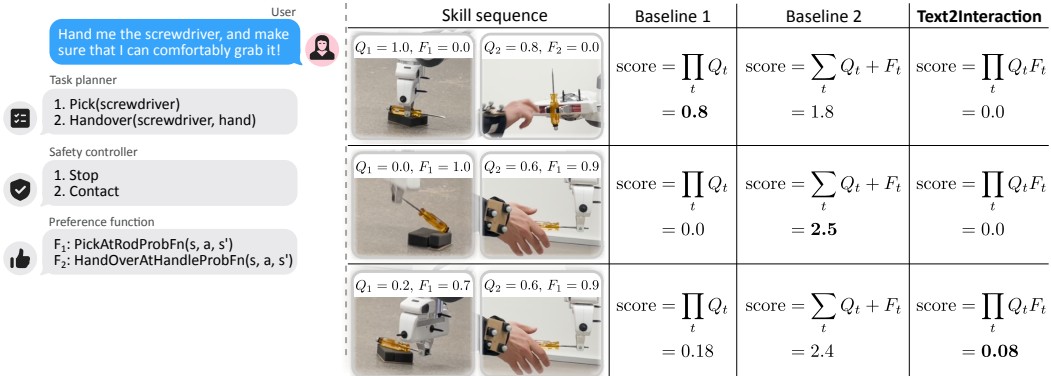

| | Skill sequence | Baseline 1 | Baseline 2 | **Text2Interaction** |
|---|---|---|---|---|
| $Q_1 = 1.0, F_1 = 0.0$  $Q_2 = 0.8, F_2 = 0.0$ | | $\text{score} = \prod_t Q_t$ $= \mathbf{0.8}$ | $\text{score} = \sum_t Q_t + F_t$ $= 1.8$ | $\text{score} = \prod_t Q_t F_t$ $= 0.0$ |
| $Q_1 = 0.0, F_1 = 1.0$  $Q_2 = 0.6, F_1 = 0.9$ | | $\text{score} = \prod_t Q_t$ $= 0.0$ | $\text{score} = \sum_t Q_t + F_t$ $= \mathbf{2.5}$ | $\text{score} = \prod_t Q_t F_t$ $= 0.0$ |
| $Q_1 = 0.2, F_1 = 0.7$  $Q_2 = 0.6, F_1 = 0.9$ | | $\text{score} = \prod_t Q_t$ $= 0.18$ | $\text{score} = \sum_t Q_t + F_t$ $= 2.4$ | $\text{score} = \prod_t Q_t F_t$ $= \mathbf{0.08}$ |

Left panel:

**User:** Hand me the screwdriver, and make sure that I can comfortably grab it!

**Task planner**
1. Pick(screwdriver)
2. Handover(screwdriver, hand)

**Safety controller**
1. Stop
2. Contact

**Preference function**
$F_1$: PickAtRodProbFn(s, a, s')
$F_2$: HandOverAtHandleProbFn(s, a, s')

Figure 8: **Preference-aligned planning with Text2Interaction.** The user asks the robot to hand them the screwdriver so that they can comfortably grasp it. Previous methods would only output a task plan and optimize it for task success (baseline 1 [5]), or treat preference as an additive reward, leading to unsuccessful executions (baseline 2 [13–16]). Text2Interaction takes task-, motion- and control-level preferences into account. In this example, our system outputs two preference functions that evaluate if the screwdriver is picked by the rod and if it is then handed over by the handle, encouraging plans toward their satisfaction. The skill sequence in the first row is the most likely to succeed, but not preferable as the handle is not easily graspable. The second sequence includes an action that is most likely going to fail, as the robot would try to grasp the very tip of the screwdriver. The last row depicts a skill sequence that is preferable and executable. From these three skill sequences, baseline 1 would select the first sequence, as it is the most likely to succeed, and baseline 2 would select the second sequence, as it maximizes the sum over $Q$-function and preference functions. Only Text2Interaction would correctly select the third skill sequence.

motion preference in each individual skill of the sequence resulting in the events

$$\forall t = 1, \ldots, H : S_{\text{motion}}^t : \text{the action } a_t \text{ satisfies the motion preferences at time step } t \text{ starting in } s_t.$$

Hereby, we assume that the motion preference $S_{\text{motion}}^t$ does not depend on the future environment history $(s_{t+1:H+1}, \psi_{t+1:H})$. This assumption was reasonable in our experiments, but there might be more complex long-horizon dependencies for which this assumption fails. The overall motion preferences of the user are only fulfilled if their motion preferences at each time step are fulfilled, i.e., $S_{\text{motion}} = S_{\text{motion}}^{1:H} = S_{\text{motion}}^1 \wedge \cdots \wedge S_{\text{motion}}^H$. We give a more detailed explanation of the steps and approximations involved in (9) following the equation.

$$p(S_{\text{motion}} \mid s_1, a_{1:H}, \underbrace{i, \phi_{1:H}, \alpha_{1:H}, \xi_{1:H}, S_{\text{feasible}}, S_{\text{task}}, S_{\text{control}}}_{\kappa}) \tag{9a}$$

$$= p\left(S_{\text{motion}}^1 \mid s_1, a_{1:H}, \kappa\right) p\left(S_{\text{motion}}^{2:H} \mid s_1, a_{1:H}, S_{\text{motion}}^1, \kappa\right) \tag{9b}$$

$$= p\left(S_{\text{motion}}^1 \mid s_1, a_1, \kappa\right) \int_{\mathcal{S}} p\left(S_{\text{motion}}^{2:H} \mid s_2, s_1, a_{1:H}, S_{\text{motion}}^1, \kappa\right) p\left(s_2 \mid s_1, a_1, S_{\text{motion}}^1, \kappa\right) ds_2 \tag{9c}$$

$$= p\left(S_{\text{motion}}^1 \mid s_1, a_1, \kappa\right) E_{s_2}\left[p\left(S_{\text{motion}}^{2:H} \mid s_2, a_{2:H}, S_{\text{motion}}^1, \kappa\right)\right] \tag{9d}$$

$$\overset{s_2 \sim T_1(\cdot \mid s_1, \psi_1)}{\approx} p\left(S_{\text{motion}}^1 \mid s_1, a_1, \kappa\right) p\left(S_{\text{motion}}^{2:H} \mid s_2, a_{2:H}, S_{\text{motion}}^1, \kappa\right) \tag{9e}$$

Repeat steps (9b)-(9e) with $s_{t+1} \sim T_t(\cdot \mid s_t, \psi_t)$:

$$\approx p\left(S_{\text{motion}}^1 \mid s_1, a_1, \kappa\right) p\left(S_{\text{motion}}^2 \mid s_2, a_2, S_{\text{motion}}^1, \kappa\right) \cdot \ldots \cdot p\left(S_{\text{motion}}^H \mid s_H, a_H, S_{\text{motion}}^{1:H-1}, \kappa\right) \tag{9f}$$

$$= \prod_{t=1}^H p\left(S_{\text{motion}}^t \mid s_t, a_t, S_{\text{motion}}^{1:t-1}, \kappa\right) \tag{9g}$$

$$\approx \prod_{t=1}^H F_t(s_t, a_t), \tag{9h}$$

where $E_{s_2}[\cdot]$ refers to the expected value over all possible states $s_2$. Note, that in (9d), the expected value over all possible states $s_2$ is conditioned on the action $a_1$, state $s_1$ and on the events

that the execution of primitive $\psi_1$ is feasible and preferable. To derive (9c), we use the assumption that $S^1_{\text{motion}}$ is independent of $a_{2:H}$. For (9e) we use the Markov assumption. To retrieve the estimate in (9f), we follow the proposed procedure in [5, Appendix C.1], where we compute a single Monte-Carlo sample estimate of (9e) under the state transition $T_1\left(s_2 \mid s_1, \psi_1\right)$. Here, the key insight is that we only execute skill sequences if the problem in (6) finds a valid action sequence that fulfills the user preferences and is feasible. Therefore, it is reasonable to assume that the condition events $S^1_{\text{motion}}$ and $S^1_{\text{feasibility}}$ in $p\left(s_2 \mid s_1, a_1, S^1_{\text{motion}}, \kappa\right)$ occurred. The approximation $E_{s_2}\left[p\left(S^{2:H}_{\text{motion}} \mid s_2, a_{2:H}, S^1_{\text{motion}}, \kappa\right)\right] \approx p\left(S^{2:H}_{\text{motion}} \mid s_2, a_{2:H}, S^1_{\text{motion}}, \kappa\right)$ was reasonable in our experiments as the learned transition distributions had a low variance. We then repeat steps (9b)-(9e) to get the approximation in (9f), which can be written as (9g). Finally, for (9h) we assume that the probability functions $F_t(s, a)$ returned by the LLM approximate the true underlying probabilities of user satisfaction by the given actions. In our ablation study, we saw that this assumption held if the new task was semantically in distribution [66] of the given in-context examples. For example, if we give Text2Interaction the screwdriver handover as an in-context example, it produces reasonable preference functions for the request "Hand me an ice cream", but it fails for the request "fold the piece of paper to a square".

