# OpenReview forum: "Text2Interaction: Establishing Safe and Preferable Human-Robot Interaction"
_robot-learning.org/CoRL/2024/Conference — CoRL 2024_

### Official Review · Reviewer_VQtk · 2024-07-20
**LLMs to improve alignment of robot behavior with human preferences. Requires better justification of claims.**

**Originality:** 4
**Technical Quality:** 3
**Clarity Of Presentation:** 3
**Potential Impact:** 3
**Recommendation:** 2
**Confidence:** 4

**Review:**

The paper identifies the need for adaptation to preferences at different levels of abstraction of robot actions, i.e. task-level, motion-level and control-level. It clearly breaks down the formulation of the role of human preferences in combined task and motion planning and control, and how the joint optimization can be inserted into this mathematical formulation. The authors identify the ability of an LLM to elaborate on human preferences beyond what is explicitly expressed in the instruction, and leverage it to optimize robot behavior to follow human preferences. However, I have the following concerns:

- Regarding the breakdown of Task (correct plan) + Motion (desired way) + Control (safe and comfortable), and it’s adaptation in the proposed method:
     - The claim that motion preferences are equivalent to ‘desired way’, and control preferences are equivalent to ‘safety and comfort’ is not well-substantiated; in that preference and safety can play a role in both of those abstractions. For instance the desire to have the robot perform quick control actions is a preference and not necessarily a safety or comfort concern, and avoiding action sequences where the pointed end of a tool faces the user is a safety concern.
     - The breakdown in eq (5) includes the task completion component, as well as three preference components for task, motion and control. Later in Section 3.2, the task completion term  (represented using Q), and one of the three preference terms (the motion term, represented using F) are used to obtain the optimization criterion. It is unclear if the other two terms dropped, or somehow implicitly included in F. In general, it is unclear how control preferences are included as a part of this formulation, especially since F only depends on the trajectory and not the control input.

- Sum of rewards v.s. Task completion probability
     - The claim that by multiplying, as opposed to adding, the task completion score (Q) and human preference score (F) results in a completely different underlying mathematics is not properly substantiated. Especially so because the numbers Q and F are a result of learned networks and not probabilities by design. For instance, one could suppose that instead of probabilities, Q and F are log probabilities, in which case Q+F would result in maximizing the combined probability as well.
     - In this particular formulation of F, the preferences are constrained to be expressed as a probability through the monotonic functions and logical functions defined in section 3.1, but there is no strong evidence to claim that this formulation would generalize beyond such a setting.

**Edit after rebuttal**: The authors' clarifications including the Q-function, details regarding the controller, the distinction between the three levels, and the sum v.s. product were helpful. However, my concerns on the minimal distinction between sum and product remain. In addition, the simplification to using constant probability terms for two of the three levels, when the main story is adapting at multiple levels of hierarchy seems like a major one. My overall recommendation remains the same.

**Quality Of The Limitations Section:**

3

**Questions For Rebuttal:**

In context of the above review:
- How are preferences for each of three levels of abstraction are/are not taken into account?
- How does the distinction between sum of rewards vs product of likelihoods stand for tasks beyond the evaluation setting?

Other clarification questions and suggestions:
- Since a large number of symbols are required to formulate the system, a block diagram annotated with the symbols would help serve as reference in parsing and tying together sections from notation to methodology.
- Does the Q function only represent geometric feasibility only, or does it also take into account the task goal? Section 3.2 claims that geometric feasibility can be approximated as the product of Q-functions over the trajectory, but if it is trained on the target task, does it also account for task completion probability in addition to geometric feasibility?
- The role of controllers in the overall system is unclear. The initial narrative suggests that the aim of the controller is to follow a trajectory, and can be parametrized by ‘eta’. But in the ‘Safe controller’ section, it is hard to connect what ‘eta’ actually constitutes. Are the control modes the ‘eta’? Relatedly, in the results of H1, it is unclear what ‘have stated a preference in the safety shield mode’ means.

**Robotics Focus:**

4

**Summary Of Paper:**

The authors propose a method to improve the way in which human preferences are accounted for in robot planning, in addition to task completion. The proposed method jointly optimizes for human preferences at each of the three levels of abstraction: task planning, motion planning and control. The idea is that the task preferences account for finding the correct plan, the motion preferences for finding the desired way, and control preferences to enable a feeling of safety and comfort. The key to this method is using an LLM to translate a human instruction into a more detailed reward function, which can be used to optimize against. Through a user study, the authors show that Text2Interaction outperforms baselines in satisfying user preferences, without a decrease in overall task completion success.

**Summary Of Recommendation:**

The paper proposes a novel method to better align robot behavior to human preferences by relying on LLMs to detail such preferences from human instructions. The paper requires more theoretical or empirical justification to support two claims: first, how the sum v.s. product formulations are fundamentally distinct, and second, how the preferences at all three levels of abstraction are accounted for.

---

### Official Review · Reviewer_BygU · 2024-07-21
**A well-researched paper with a novel approach but needs refinement in presentation of the proposed method and experiments.**

**Originality:** 3
**Technical Quality:** 3
**Clarity Of Presentation:** 1
**Potential Impact:** 3
**Recommendation:** 3
**Confidence:** 5

**Review:**

The paper presents a novel approach to integrating human preferences into robotic systems using LLMs. The use of LLMs to generate not only task plans but also motion preferences as Python code, and controller parameters for safety is well-motivated and demonstrates potential for integrating human preferences into robotics.

Strengths:

1. The approach addresses both task-level and motion-level preferences, which is a novel aspect compared to existing methods that often focus on one level.
2. The experiments include real-world user study with statistically significant results that provide empirical evidence of the proposed method.
3. The paper addresses a key challenge in human-robot interaction—adapting to individual user preferences quickly and efficiently. The paper presents a promising approach, demonstrating its potential in integrating new user preferences in a zero-shot manner.

Weaknesses:

1. OR operation is wrong in section 3.1. I believe $OR(p_1, p_2)$ should be calculated as $p_1p_2 + p_1(1-p_2) + p_2(1-p_1) = p_1+p_2-p_1p_2$, not $max(p_1, p_2)$, where the two events are stochastically independent. How does this error affect the proposed algorithm and its performance, or any of the derivations?
2. There exist numerous recent work using LLMs to output parameters on robot control (oftentimes reward function or its parameters) to align with human preferences given as language instructions. Citing previous works such as [1, 2, 3] and comparing them with the proposed method would be essential.
3. Although the paper is well-structured, the paper's clarity can be significantly improved, especially in the method section. Simplifying the notations and providing more intuitive explanations would help.
4. While the user study and ablation study provide valuable insights, additional experiments with a larger and more diverse set of tasks would strengthen the validation of the approach.
5. Addressing the terminologies clearly would help readers understand the paper easily, even though the paper builds its formulation on top of Text2Motion [4]. For instance, what does $S_{task}(\phi)$ refer to in L189? Is $H$ in equation 1 time horizon? Describing what 'controller parameters' mean in section 1.2 or 1.3 or having examples would improve the readability of the paper since its meaning appears later in the experiments section for the first time.
6. As mentioned in the limitations, the proposed framework's reliance on LLMs raises questions about its robustness and generalizability to diverse and unexpected user instructions.
7. There are some typos - L38 (optimizes -> optimize) and L200 (investigated -> investigate).

**Quality Of The Limitations Section:**

3

**Questions For Rebuttal:**

1. Can the authors provide more detailed examples of how the preference functions generated by the LLMs align with user instructions?
2. When does the shooting objective become zero? Could the authors report numbers on how often greedy-search happens and what those mean?
3. Could the authors elaborate on the computational efficiency of Text2Interaction, particularly in real-time applications?

[1] Xie, Tianbao, et al. "Text2Reward: Reward Shaping with Language Models for Reinforcement Learning." The Twelfth International Conference on Learning Representations.

[2] Ma, Yecheng Jason, et al. "Eureka: Human-level reward design via coding large language models." arXiv preprint arXiv:2310.12931 (2023).

[3] Hwang, Minyoung, et al. "Promptable behaviors: Personalizing multi-objective rewards from human preferences." Proceedings of the IEEE/CVF Conference on Computer Vision and Pattern Recognition. 2024.

[4] K. Lin, C. Agia, T. Migimatsu, M. Pavone, and J. Bohg. Text2Motion: From natural language instructions to feasible plans. Autonomous Robots, 2023

**Update after rebuttal:**
I have carefully read the authors' rebuttal and, in particular, the answers to the questions I raised. I am glad that the authors have experimented with the correct OR implementation and also fixed minor errors in the paper.

**Robotics Focus:**

4

**Summary Of Paper:**

The paper proposes a framework named Text2Interaction, which aims to integrate human preferences into robot task planning, motion planning, and control in a seamless manner. The approach leverages large language models (LLMs) to generate task plans, motion preferences as Python code, and parameters for a safe controller. By optimizing for both task completion and user satisfaction, Text2Interaction aims to provide a more adaptable and responsive human-robot interaction. The framework is validated through a user study on real robots and an ablation study in simulation, showing promising results in preference integration and user satisfaction.

**Summary Of Recommendation:**

This paper presents a novel framework for integrating human preferences into robot task planning, motion planning, and control using LLMs. While the approach is shows promising results, the paper needs clearer explanations on the formulation of the proposed method and the simulation experiments.

---

### Official Review · Reviewer_dJ5U · 2024-07-31
**A new method that integrates user preference in task and motion planning**

**Originality:** 4
**Technical Quality:** 3
**Clarity Of Presentation:** 5
**Potential Impact:** 3
**Recommendation:** 3
**Confidence:** 3

**Review:**

# Summary

The authors propose *Text2Interaction*, a framework integrating human preferences in task, motion, and control planning of robots.

## Method
This method prompts LLMs to generate user preference function $F_t(s_t, a_t)$ as Python code with a list of building block functions, e.g. pose, metric, and probability functions. This function is multiplied by Q function and optimize the trajectory with shooting method from Text2Motion. If shooting fails, the method falls back to a greedy search that optimizes $S_{task}F_tQ_t$

## Experiments
With experiments, the authors tested two scenarios: user study and simulation ablation.
### User study: Franka pick up and hand over task.
The authors set up three stages:
1. User choose the reactive mode;
2. The robot performs based on $\arg\max\Pi Q_t$ w/ predefined instruction
3. The robot performs based on $\arg\max\Pi Q_tF_t$ w/ predefined instruction
4. 3 but w/ personal instruction

The questionnaire results demonstrated 83% of users agree Text2Interaction integrates their preferences, 94% prefer it over the baseline.

### Simulation: Rearrangement
On 10 instructions, the authors compare their method also with groundtruth hand-crafted $QF^*$ and $Q + F$, and showed that their method is more preferable against $Q+F$ and a headroom towards using $F^*$.

# Strength
1. The idea of generating optimization method for human preference is pretty novel and could work for a variety of natural language expressions.
2. The experiments are convincing for the scenarios that the authors designed. But this could be hard to generalize as will be mention later.


# Weakness
1. User study is only performed on one single scenario. It is better to consider diverse setups.
2. Related to 1, the preference function is limited to the primitives in Fig. 5. I am not sure if this is generalizable for open domain tasks.

**Quality Of The Limitations Section:**

2

**Questions For Rebuttal:**

1. I am confused about the concrete implementation of $S_{task}$ and why *sum of token log-probabilities of each primitive’s
language description* is a good optimization objective?
2. I might have missed this, but why didn't you compare with $\Sigma Q + F$ in user study? And would $\Sigma Q+F$ be different from $\Pi Q+F$?
3. How could user preference describing motion be expressed? For example, *put the vase on the desk carefully*. This may be out of scope of Fig. 5

**Robotics Focus:**

4

**Summary Of Paper:**

Text2Interaction is a method that outputs user preference function, which is used as a multiplier for optimization objective to integrate user preference into task and motion planning.

**Summary Of Recommendation:**

This paper has its technical merit for the design that makes user preference easily integrates into planning. This is useful for HRI research community. The major concern is around the generalizability: how would this be applied to other settings including different scenarios, and user preference that cannot be expressed through the building block functions.

---

### Author Rebuttal · Authors · 2024-08-08

We sincerely thank the reviewers for their invaluable feedback and in-depth reviews of our manuscript.
We are particularly grateful for the acknowledgement of our manuscript's originality and significance in advancing the development of general-purpose human-robot interaction models, which reaffirms our motivation and belief in our research's potential.
In our rebuttal file, we provide our reworked manuscript with text highlights of the suggested changes and full answers to all reviewers in PDF format.

## Main changes
1. Adjusted the definition of motion and control preferences in the introduction
2. Reworked safe controller section to give a more intuitive understanding and moved the reachability analysis formulation to the appendix.
3. Improved methodology section by adding more details to Fig. 2 and rephrasing key definitions.
4. Added a section to the appendix that gives practical examples on how Text2Interaction integrates human preference.

---

### Decision · Program_Chairs · 2024-09-04

**Decision:**

Accept

**Comment:**

Summary of the Paper

The paper presents "Text2Interaction," a framework designed to seamlessly integrate human preferences into robot task planning, motion planning, and control. This approach employs large language models (LLMs) to generate task plans, motion preferences in Python code, and safe controller parameters. By maximizing the combined probability of task completion and user satisfaction, Text2Interaction aims to provide adaptable and responsive human-robot interaction. The framework is validated through both a real-world user study and an ablation study in simulation, which show promising results in user satisfaction and preference integration.

Strengths
- The method of using LLMs to integrate human preferences across task, motion, and control levels is novel and represents a major advance over existing methods that typically focus on a single aspect.
- The framework is evaluated through a combination of real-world user studies and simulation-based ablation studies, providing a strong empirical foundation. The results demonstrate statistically significant improvements in user satisfaction and preference integration.
- The ability of Text2Interaction to react to new user preferences in a zero-shot manner highlights its potential for adaptable and flexible human-robot interaction, which is a significant challenge in the field.
- The framework places a strong emphasis on aligning with human preferences, which is crucial for enhancing the efficacy and acceptance of robots in various applications.

Weaknesses
- The user study was conducted with a single task scenario, raising concerns about the generalizability of the approach across diverse setups and tasks. Additional experiments with a wider variety of tasks would strengthen the validation.
- While the paper is technically sound, the presentation, particularly in the methodology section, could be clearer. The paper employs complex notations and lacks intuitive explanations, which may hinder understanding.
- There is a lack of theoretical or empirical justification for certain claims, particularly regarding the distinctiveness of the sum versus product formulations in optimizing task completion and human preference scores. This needs further exploration.
- The preference functions are constrained by the primitives available, which may limit the framework's applicability to open-domain tasks or preferences not easily expressed by the predefined building blocks.
- The reliance on LLMs raises questions about robustness and generalizability to diverse and unexpected user instructions, which could impact the framework's applicability in dynamic real-world settings.

Summary of the rebuttal phase

While the authors' revisions appear to have improved the quality of the paper, the rebuttal did not change the reviewers' opinions. Although not all reviewers agree, accepting as a poster presentation is considered reasonable.